# ON THE EVALUATION OF GENERATIVE ROBOTIC SIMULATIONS

## ABSTRACT

Due to the difficulty of acquiring extensive real-world data, robot simulation has become crucial for parallel training and sim-to-real transfer, highlighting the importance of scalable simulated robotic tasks. Foundation models have demonstrated impressive capacities in autonomously generating feasible robotic tasks. However, this new paradigm underscores the challenge of adequately evaluating these autonomously generated tasks. To address this, we propose a comprehensive evaluation framework tailored to generative simulations. Our framework segments evaluation into three core aspects: *quality*, *diversity*, and *generalization*. For single-task quality, we evaluate the realism of the generated task and the completeness of the generated trajectories using large language models and vision-language models. In terms of diversity, we measure both task and data diversity through text similarity of task descriptions and world model loss trained on collected task trajectories. For task-level generalization, we assess the zero-shot generalization ability on unseen tasks of a policy trained with multiple generated tasks. Experiments conducted on three representative task generation pipelines demonstrate that the results from our framework are highly consistent with human evaluations, confirming the feasibility and validity of our approach. The findings reveal that while metrics of quality and diversity can be achieved through certain methods, no single approach excels across all metrics, suggesting a need for greater focus on balancing these different metrics. Additionally, our analysis further highlights the common challenge of low generalization capability faced by current works. Our anonymous website: https://sites.google.com/view/evaltasks.

## 1 INTRODUCTION

Embodied artificial intelligence (EAI) is crucial to enable intelligent agents to understand and interact with the physical world. However, creating such agents with physical forms and universal functionalities necessitates extensive data, which is prohibitively expensive to acquire manually (Dasari et al., 2020; Srivastava et al., 2021; Mu et al., 2021). Although multiple attempts have been made toward massive real-world data collection (Brohan et al., 2023b;a), training in simulated environments still plays a key role in various robotic tasks (Wang et al., 2023a; Huang et al., 2021; Lin et al., 2021; Yuan et al., 2024; Yu et al., 2020). Consequently, the acquisition of a substantial number of robotic tasks in simulation, which heavily rely on foundation models, is of significant importance.

Foundation models (OpenAI, 2023; Team et al., 2023; Zhang et al., 2023a) have exhibited remarkable proficiency in various robotics-related tasks, including coding (Rozière et al., 2023), 3D generation(Deitke et al., 2022; 2023), scene comprehension (Mohiuddin et al., 2024), planning (Huang et al., 2023b; 2024), and reward formulation (Ma et al., 2023). Notably, recent works have demonstrated the potential of leveraging such capabilities of foundation models to generate robotic tasks in simulation (Wang et al., 2023b; 2024; Katara et al., 2023; Yang et al., 2024; Hua et al., 2024). In generative simulation, foundation models such as large language models and vision-language models are prompted to output necessary task information (e.g., code, language descriptions), an appropriate scene, and successful trajectories for novel tasks at scale. However, despite these advancements, concerns have been raised regarding aspects such as the quality and reality of the generated tasks and whether the generated data can boost policy performance (Hua et al., 2024). Therefore, there is an urgent need for a comprehensive evaluation framework for generative simulation pipelines, which has so far been absent.

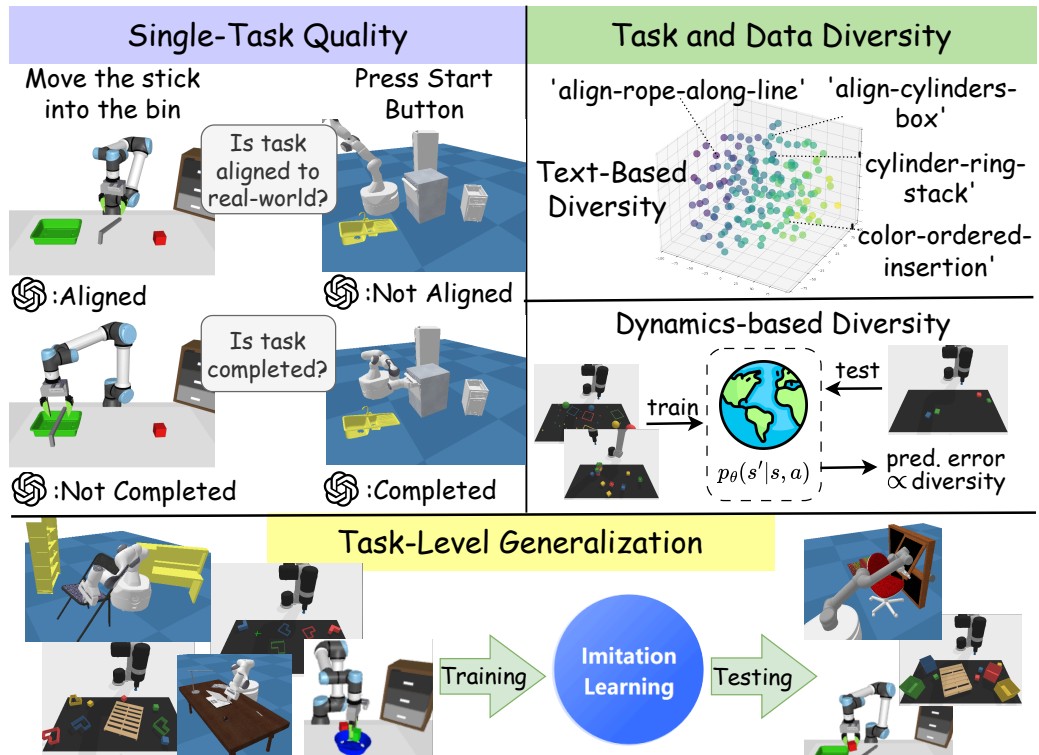

Figure 1: We propose three main aspects for evaluating generative simulations: **Quality, Diversity, and Generalization**. Quality encompasses two components: the alignment of the task scene with the real world, and the completion score, which assesses if the robot's trajectory solves the task. Diversity is divided into two components as well: text-based diversity of task descriptions, and dynamics-based diversity among trajectory data. Generalization involves assessing the data's generalization ability using a representative imitation learning model.

However, evaluating the generated tasks faces challenges similar to those encountered in assessing images (Salimans et al., 2016; Heusel et al., 2017) and texts (Wang et al., 2018), i.e., it is hard to quantify the realism of the generated tasks, thus hindering traditional evaluation mechanisms such as success rate from reflecting the quality and value of the generated tasks and data. In this paper, we propose a novel evaluation framework (see Fig 1) tailored to generative simulation pipelines. Our framework is concerned with three key perspectives: (1) the *quality* of single-task generation, which typically involves the alignment score of generated task scenarios with the real world and the completeness of generated task trajectories; (2) the task and data *diversity* concerned with generated tasks as well as generated trajectories; (3) the task-level *generalization* ability of a policy trained on a bunch of generated tasks.

Specifically, in terms of single-task quality, we leverage vision-language models to understand scene/trajectory images and output the scene alignment scores and task completion rates, which have been measured by human subjective judgment or hard-coded functions in previous works. Besides thorough single-task evaluation, we also incorporate multi-task evaluation on diversity and generalization, which are not extensively investigated in prior studies. For diversity, we first measure task diversity by examining the text similarity between the language descriptions of generated tasks. We then train a world model with trajectory data collected from these tasks and evaluate trajectory diversity based on the model's prediction loss. For generalization, we train an imitation learning policy based on a variety of generated tasks and measure its efficacy on unseen tasks to gauge its task-level generalization capability.

In our experiments, we study 3 notable projects, namely, GenSim (Wang et al., 2024), RoboGen (Wang et al., 2023b), and BBSEA (Yang et al., 2024), in the hope of establishing a reference for subsequent research towards this direction. By comparing our evaluations with those from humans, we find that our evaluations reach consistent conclusions with the human experts on the vast ma-

jority of tasks. According to our evaluation results, RoboGen's tasks exhibit the highest single-task quality while also holding an advantageous position in the textual task diversity of task descriptions. In terms of trajectory diversity, both GenSim and BBSEA have demonstrated superior results. Although currently none of the pipelines possess sufficiently excellent generalization capabilities, the tasks from GenSim still show a certain degree of potential in the direction of generalization. These findings indicate that although specific methods can achieve satisfactory quality and diversity metrics, none consistently outperform across all criteria, emphasizing the need for intensified efforts to balance these metrics effectively. Furthermore, our study also emphasizes the prevalent issue of low generalization capability encountered by current methodologies.

The main contributions of our work can be summarized as follows:

- We propose a novel framework to evaluate generative robotic simulation methods, providing researchers with tools to assess and improve their future works in this area.

- We develop an autonomous pipeline that quantitatively assesses the quality of an individual task using foundation models, which have previously been performed by human efforts.

- We introduce metrics for diversity and generalization of generated tasks and data to evaluate the value of multiple generated tasks and the extensive data derived from them.

## 2 RELATED WORKS

### 2.1 FOUNDATION MODELS

In our article, we utilize large language models such as GPT-4 (OpenAI, 2023), released by OpenAI, which have had a profound impact on the field of natural language processing. Previous work has applied these large language models to the domain of robotics, specifically in policy learning (Driess et al., 2023; Huang et al., 2023c) and motion planning (Huang et al., 2022). Researchers have also explored using language models to generate code and rewards (Huang et al., 2023c; Wang et al., 2023b), aiding solvers in learning policies from tasks. Furthermore, vision-language models and multimodal foundational models have demonstrated remarkable potential (Zhang et al., 2023a; Team et al., 2023; Xu et al., 2023). Model GPT-4-vision (Zhang et al., 2023a) have exhibited capabilities in spatial understanding and basic assessment, making the automatic evaluation of tasks feasible. In prior work, vision-language models have been employed in the robotic task generation pipeline to verify the quality of the generated tasks (Wang et al., 2023b; 2024).

### 2.2 GENERATIVE ROBOTICS TASKS AND DATASETS IN EMBODIED AI

In recent research, foundational models have demonstrated remarkable capabilities (OpenAI, 2023; Zhang et al., 2023a; Team et al., 2023), leading to the emergence of autonomously generated robotic tasks in the field of robotics. Typically, such generative models utilize large language models to create a basic framework for generating tasks, which involves submitting the required 3D models through text-to-3D model (Li et al., 2023b) conversion, text-to-image (Mid) and image-to-3D models (Liu et al., 2023) processes, or searching and generating the necessary three-dimensional models in extensive 3D model repositories like Objaverse (Deitke et al., 2022; 2023). These models are then assembled into tasks within simulators, and methods such as reinforcement learning or trajectory optimization are employed to learn the trajectories needed to solve the tasks. Researchers have also explored tasks in other directions; for instance, the creators of Robogen have expanded task types to include soft materials and humanoid robots (Wang et al., 2023b), while the developers of Gensim have opted to deploy tasks on real robots, completing the generated tasks in the real world (Wang et al., 2024). Therefore, when evaluating the quality of generated tasks, it is also necessary to consider the diverse directions of exploration being pursued by different researchers.

### 2.3 BENCHMARKS ON MESHES AND LARGE LANGUAGE MODEL

Recent work has bridged gaps in the evaluation of three-dimensional models and large language models. For instance, T3Bench (He et al., 2023) introduced the use of multiple foundational models to establish an evaluation system for metrics such as the quality and alignment of 3D models. The methods used in this work to assess the quality and alignment of 3D models have inspired our

approach to evaluating the alignment of task scenarios in robotic tasks. Additionally, in the evaluation of large language models (Zhang et al., 2023b; Huang et al., 2023a), previous studies have discussed assessing various metrics across multiple scenarios to identify potential issues of hallucination and errors within models. These evaluation standards provide a valuable perspective for assessing robotic tasks, aiding in a more appropriate evaluation of such tasks.

### 2.4 Evaluation of Task Diversity

Learning a range of skills is crucial for building generalist robot agents that can autonomously operate in a complex environment. Therefore, we expect task generation to produce tasks with varying goals, dynamics, and environment setups such that collectively learning these tasks promotes generalization, robustness (Tobin et al., 2017), and even unseen skills (Eysenbach et al., 2018). However, evaluating such diversity of generated tasks remain unclear. RoboGen (Wang et al., 2023b) proposed to compare the Self-BLEU score and Embedding Similarity (Zhu et al., 2018) of the descriptions generated alongside the tasks. While such language-based diversity metrics consider high-level semantic information, they are strongly coupled with the language models used, which are known to be subject to alignment issues. In this work. we propose to evaluate task diversity as the coverage of skill or dynamics space, where high diversity facilitates better transfer or generalization to a held-out set of tasks. Recent model-based skill learning methods (Hafner et al., 2023; Hansen et al., 2024) are capable of learning highly complex dynamics and task information on a wide range of tasks. We leverage them for diversity evaluation.

## 3 Method

### 3.1 Introduction to Generative Simulation

Generative simulation represents a field of studies that utilize foundation models, particularly generative models pre-trained on internet-scale data, to acquire massive tasks and data in robot simulation. In generative simulation, large language models are first prompted to provide the basic framework for a novel task, such as the task description, assets, task code, etc. The task is then loaded into the simulation to construct a scene. We further query foundation models to provide objectives for task solutions, e.g. goals for planning or rewards for RL. Through RL training or motion planning, the pipeline will produce trajectory data for the previously generated task. To summarize, the performance of a generative simulation pipeline is fundamentally determined by key aspects such as the basic task framework, solution objectives, and the specific implementations of solution generation.

### 3.2 Overview

We divide our evaluation work into three parts, as visualized in Fig 2. In the first part (Sec 3.3), we assess the quality of a single generated task through foundational models, especially large language models and vision-language models, and statistical methods. In the second part (Sec 3.4), we respectively measure the diversity of generated task descriptions and trajectory data with a language model and a world model. In the third part (Sec 3.5), we evaluate the generalization capability of an imitation learning policy distilled from a large number of generated tasks.

### 3.3 Single-Task Quality

In this section, we introduce how we evaluate single-task quality. We consider two metrics: scene alignment score which measures the realism of the generated task and task completeness score which measures whether the generated task is successfully solved to collect data.

**Scene alignment score.** We utilize two different pipelines to evaluate scene alignment score. Regarding "the realism of generated tasks", it includes whether the scene is aligned with the text, as well as the "semantics" of real scenarios. Due to possible deficiencies in visual recognition from foundation models (Tong et al., 2024b), one of our methods uses visual models, e.g., BLIP2 (Li et al., 2023a), to generate textual descriptions of rendered scene images, followed by large language models (LLMs) such as GPT-4 (OpenAI, 2023) to assess the consistency between the textual descriptions and the task descriptions. The other directly employs multi-modal LLMs like GPT-4V (Zhang et al.,

Figure 2: Overview of our evaluation framework. In our method, the evaluation is divided into three parts. We initially employ LLM and VLM to evaluate scene alignment and task completion for generated tasks. These tasks are subsequently categorized into groups for assessment on two fronts: task diversity, gauged by the textual similarity of task descriptions, and data diversity, measured by prediction errors from a world model. Finally, we assess the generalization capability of a policy trained on generated data.

2023a) and LLaVA (Liu et al., 2024) to evaluate the consistency between the task descriptions and the scene images. For complex scenes, we used multi-view images for evaluation.

**Task completion score.** We use foundation models, particularly vision-language models (VLMs), to assess the completion score of a generated task trajectory. Previously, this assessment was conducted using hard-coded functions, which demonstrated only limited capability in measuring task completion for specific tasks. Specifically, our approach begins with generating a video of the robot's trajectory as we execute the task solution. From the video, we extract 8 images and provide them, along with the task description, to a VLM. We then obtain an evaluation of the task completion status from the VLM.

To reduce possible inherent biases and instability within foundation models, we conduct multiple scoring iterations and take the mean scores when evaluating on both metrics.

## 3.4 TASK AND DATA DIVERSITY

The generated tasks are expected to be diverse so that training on these tasks grants agents a range of skills and the ability to operate in various situations. However, a concrete definition of diversity is hard: in what sense are tasks distinct or similar? In this work, we are concerned with diversity from the following perspectives: (1) task diversity, a high-level diversity as identified by LLMs; and (2) trajectory diversity, a low-level diversity in terms of the dynamics of the collected data.

**Text-based task diversity.** Since LLMs generate tasks including the task descriptions and possibly scene configurations and goals, they are supposed to have an internal understanding of diversity at a high level. For example, "stack-blocks-tower" differs from "align-balls" semantically in terms of the action (verb) and the object of interest. Therefore, the similarity between embeddings of task descriptions can be considered as the similarity between tasks. Specifically, following (Zhu et al., 2018), we compute the diversity of a task set with text embeddings $\{\mathbf{e}_i\}_{i=1}^N$ as:

$$\text{div} = -\frac{1}{N}\sum_i \log(\frac{1}{N-1}\sum_{i\neq j}\mathbf{e}_i^T\mathbf{e}_j), \tag{1}$$

where $N$ is the number of tasks and $i \neq j$ removes self-similarity. A higher value indicates lower similarity and hence higher diversity.

**Dynamics-based trajectory diversity.** Though straightforward, task description diversity itself does not sufficiently characterize the actual learning experience of tasks, e.g., different interaction dynamics will take place when training on different generated tasks. Ideally, a diverse set of tasks should cover a large space of dynamics to promote the agent's robustness under different scenarios. Therefore, we propose to evaluate such diversity through prediction error of dynamics models. Dynamics prediction error has been associated with novelty and widely adopted to promote exploration (Pathak et al., 2017; Burda et al., 2018). A high dynamics prediction error indicates unfamiliar (and thus novel) dynamics being experienced. We leverage a latent dynamics model $p_\theta(o_{t+1}|o_t, a_t)$ following DreamerV3 (Hafner et al., 2024), where $o_t$ and $a_t$ are the observation and action at time step $t$. The model is trained on trajectories collected from the generated tasks and then evaluated to compute the prediction errors. As will be discussed in Section 4.2, this approach helps us to identify tasks that render notably similar dynamics and are therefore not diverse.

### 3.5 TASK GENERALIZATION

Generalization can be an ambiguous yet vital metric for evaluating the capabilities of generalist robot agents. In this paper, we define generalization as the capability to solve tasks within the same distribution, specifically whether an agent trained on the generated tasks can address similar scenarios and objectives albeit with varying initial states and minor low-level variations. To quantitatively examine this capability, we first train an imitation learning policy with trajectories collected by either the oracle policies or policies learned from the generation pipeline. The trained policy is subsequently evaluated with new task scenarios including varied object instances, appearance, and initial poses. The policy uses the state-of-the-art algorithm called Diffusion Policy (Chi et al., 2023) as the backbone and takes as input RGB observations, and the proprioceptions. Although BAKU(Haldar et al., 2024) meets our needs, choosing the more widely known diffusion policy method for our evaluation is reasonable. A typical indicator of low generalization is when the trained policy performs well on the training data, confirming correct algorithm implementation, yet struggles to adapt to the varied tasks during evaluation.

## 4 EXPERIMENT

### 4.1 SINGLE TASK EVALUATION

**Experimental setup.** As mentioned in Section 3.3, our methodology utilizes vision-language models (VLMs) to generate scene captions, which are then compared against task descriptions using large language models (LLMs). Additionally, we employ a multi-modal LLM (MLLM) to evaluate the completeness of task trajectories. For captioning scene images, we deploy several VLMs, including BLIP ("blip2-flan-t5-xl"), Cambrian ("Cambrian-8B") (Tong et al., 2024a), and LLaVA 1.6 ("LLaVA-1.6-7B") (Liu et al., 2024). The scene alignment score is measured using the GPT-4 ("2024-02-15-preview" version from Microsoft Azure) model. For assessing task completion, the MLLM models used include GPT-4V, Cambrian, and LLaVA.

#### 4.1.1 HUMAN VERIFICATION

To validate the efficacy of our method, we gather human evaluations for ten tasks from the released tasks of RoboGen and GenSim and examine the consistency of our results with human results. We characterize their relationship by using the Pearson correlation coefficient to represent correlation strength and the mean absolute error (MAE) to indicate numerical similarity. A higher Pearson correlation coefficient signifies a stronger correlation, while a lower MAE reflects greater similarity. Therefore, we calculate the ratio of the Pearson correlation coefficient to the MAE to assess the relationship between our method and human evaluations; higher values indicate greater similarity.

In Figure 3 left, in terms of scene alignment score, the performance of architectures using GPT-4 and other vision models like BLIP2 and LLaVA is shown to yield better results for RoboGen's tasks, but these models perform poorly on GenSim's tasks, primarily due to their lack of knowledge regarding top-view rendered images. In addition, Figure 3 right illustrates that for task completion score, GPT-

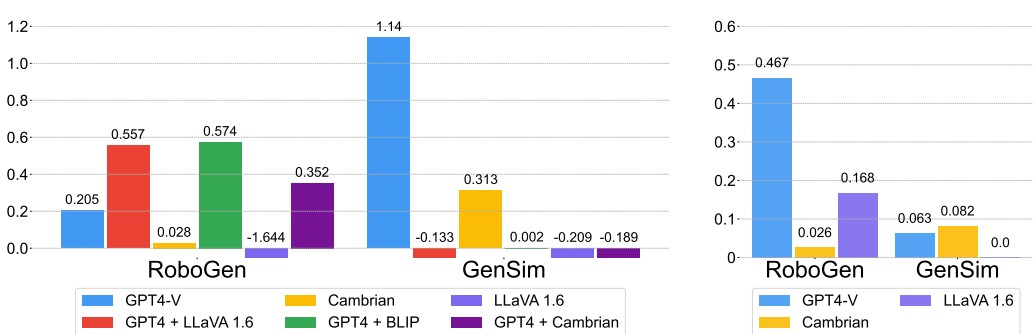

Figure 3: Pearson correlation divided by mean absolute error of the different methods with human evaluation in different datasets. In the bar chart, relatively high values indicate that the model's results are more similar to human evaluations, while negative values indicate that the model's output is negatively correlated with human evaluations. We truncate the negative bars for better visualization.

4V exhibits relative performance compared to human evaluations, indicating a strong alignment with human behavior in assessing task completion. In contrast, Cambrian and LLaVA 1.6 produce results that do not correspond with human assessments. While both models have demonstrated an understanding of images during the experiments, they fail to provide completion scores that align with human evaluations based on the image results.

### 4.1.2 EVALUATION ON ROBOGEN, GENSIM, AND BBSEA

We summarize the evaluation results of both metrics for single-task quality in Figure 4. To be specific, among the methods, RoboGen tasks demonstrate high task completion scores, but their scene alignment scores are notably low. This discrepancy arises because, although RoboGen can generate assets relevant to the current task, these assets often collide when loaded into the scene, resulting in a cluttered and difficult-to-recognize environment. In contrast, GenSim secures the highest scene alignment scores but underperforms in task completion. This shortfall is largely attributed to its vision-language model lacking access to top-view rendered data, which impairs its ability to accurately recognize task completion. In addition, BBSEA achieves decent results on both metrics (although not the best), and it has the smallest variance in the outcomes.

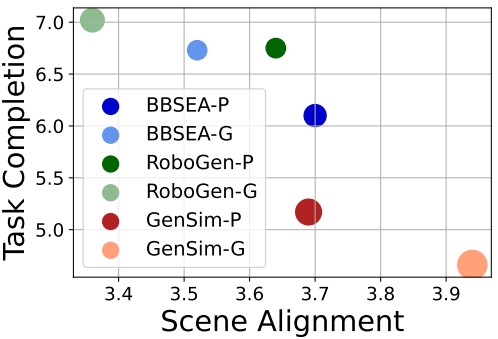

Figure 4: Single task evaluation results. "-P" flag refers to the published tasks of a certain method, while "-G" flag refers to generated tasks by running released codes. The size of the data marker represents the variance of the evaluation results under the corresponding setting.

Furthermore, we observe performance discrepancies between published and newly generated tasks across all methods. While a predictable decline in performance for generated tasks can be attributed to additional filtering prior to project release, improvements have been noted in GenSim's scene alignment and task completion for RoboGen and BBSEA. The underlying reason is the advancements in the performance of foundation models, which have expanded the limits of task generation quality, including reasonable solution objectives in RoboGen and BBSEA, and innovative long-horizon task proposals in GenSim.

### 4.1.3 EXAMPLES FROM EVALUATION

As shown in Figure 5, in the snapshot of the 'Open Laptop' task, the laptop is correctly placed on the table, and there are some objects such as a lamp and a pen placed on the desk. Then we can observe from three trajectory images that the robotic arm has correctly located the laptop and opened it. Therefore, this task gets an average score of '7.96' (out of 10) for completion score and an average

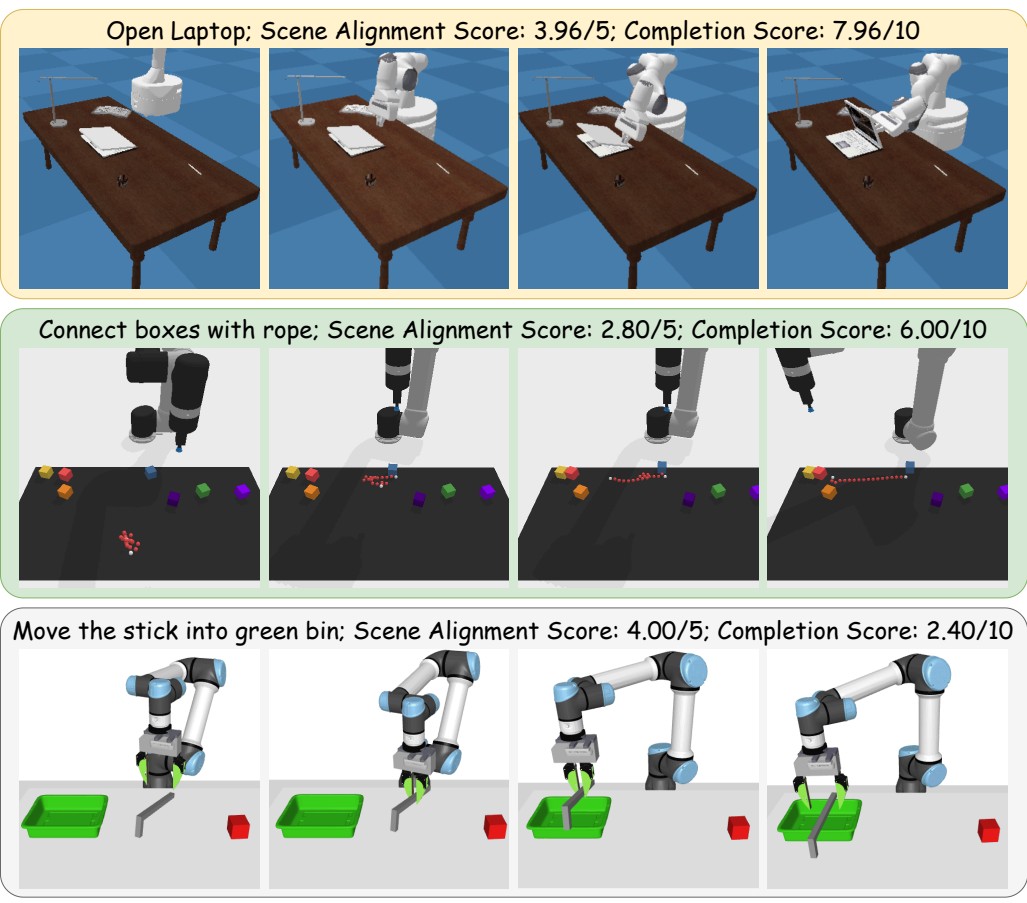

Figure 5: Single-task evaluation examples on three different tasks from different generative simulation pipelines. The first row displays a task that achieves high scores in both scene alignment and task completion. The second row illustrates a task with low scene alignment, while the third row presents a task with low task completion.

score of '3.96' (out of 5) for scene alignment. In the snapshot of the 'Connect boxes with rope', although we can abstract the red balls into a rope, there is also a gap between the scene and the real world, which gets '2.80' in scene alignment. But when the red balls are expanded into a line, our pipeline can correctly figure out the task has been solved, thus receiving a '6.00' completion score. In the snapshot of the 'Move the stick into the green bin', the gripper has grasped the stick and put it on the green bin rather than into the green bin, therefore our pipeline grades the task completion with '2.40'. But the scene receives '4.00' because there are necessary objects, as well as some relevant ones on the table in the scene.

## 4.2 TASK AND DATA DIVERSITY

In this section, we use the proposed evaluation protocols to examine whether a pipeline generates diverse tasks and hence diverse data for learning. To have a better perspective for analysis and allow practical training, we divide the tasks into groups according to skills, scene configuration, or objects involved. The details for grouping can be found in Appendix A.3.

**Task diversity.** For text-based task diversity, we leverage different language models, including "MiniLM-L6-v2" and "Mpnet-base-v2" from SentenceTransformers (Reimers & Gurevych, 2019; 2020), "LLama-3.1-70B" and "LLama-3.2-90B" (Touvron et al., 2023; Dubey et al., 2024), to generate text embeddings from task descriptions. The diversity is then measured by embedding similarity between and among the template and generated tasks following Equation (1). We consider different task groups as well as the whole task set for each method. The results are listed in Table 1, with

Table 1: Results for text-based task description diversity. The evaluation considers various task groups as well as the entire task set (**All**) for all three methods. Higher values indicate higher diversity.

| Method | Task Group | MiniLM-L6-v2 | Mpnet-base-v2 | LLama-3.1-70B | LLama-3.2-90B |
|---|---|---|---|---|---|
| GenSim | Stacking | 0.49 | 0.48 | 0.23 | 0.33 |
| | Placement | 0.55 | 0.52 | 0.12 | 0.15 |
| | Piles | 0.42 | 0.45 | 0.26 | 0.40 |
| | Assembling | 0.53 | 0.44 | 0.20 | 0.29 |
| | **All** | **0.75** | **0.70** | **0.25** | **0.34** |
| RoboGen | Table-Top | 0.88 | 0.71 | 0.22 | 0.26 |
| | Ground | 0.78 | 0.65 | 0.24 | 0.24 |
| | **All** | **0.84** | **0.69** | **0.25** | **0.28** |
| BBSEA | Table-Top | 1.22 | 1.06 | 0.24 | 0.26 |
| | Drawer | 0.37 | 0.34 | 0.27 | 0.28 |
| | **All** | **1.28** | **1.10** | **0.28** | **0.29** |

Table 2: Results for dynamics-based trajectory diversity. World model evaluation error is reported with a different number of training episodes. For a diverse task group, the prediction error should drop as the training episodes increase.

| Method | Task Group | Eval error on 10 ep | Eval error on 20 ep | Eval error on 40 ep |
|---|---|---|---|---|
| GenSim | Stacking | 115.0 | 67.2 | 24.5 |
| | Placement | 245.0 | 177.6 | 29.5 |
| | Piles | 68.3 | 47.6 | 14.5 |
| | Assembling | 105.4 | 64.3 | 29.6 |
| RoboGen | Table-Top | 16.2 | 10.9 | 6.4 |
| | Ground | 45.8 | 28.6 | 14.5 |
| BBSEA | Drawer | 402.4 | 287.0 | 69.6 |
| | Table-Top | 561.6 | 296.6 | 87.9 |

higher values indicating higher diversity. Among the three methods, BBSEA shows the highest task description diversity by our proposed metric (1). However, we observe that many drawer tasks share remarkably similar descriptions, e.g., "open the drawer using handle". Accordingly, the diversity of table-top tasks is significantly higher than that of drawer tasks. Conversely, results in GenSim indicate low task diversity because GenSim only deals with table-top pick-place tasks, narrowing its task domain. In addition, despite the extensive task types and complicated scenes RoboGen can support, it acquires lower scores than BBSEA. We attribute this underperformance to some vague task descriptions generated by RoboGen, which hinders the text similarity from reflecting task diversity.

Moreover, for all methods, we observe notable inconsistency between the language models from which we obtain the embeddings, possibly due to the difference in training method and objective that make the models attend to different components of the descriptions. This suggests that despite simplicity, text evaluation does not consistently and reliably capture the diversity of generated tasks.

**Trajectory diversity.** In terms of dynamics-based trajectory diversity, we examine whether the generated tasks provide trajectories with diverse dynamics coverage with a world model. Specifically, a total of 40 episodes is collected for each task in a group using the policy for each method. We train a world model in DreamerV3 using 10, 20, and 40 episodes respectively, and evaluate all 40 episodes. Please refer to Appendix A for details. Intuitively, a task group with low diversity is likely to exhibit *comparably low prediction* errors across various numbers of episodes, as a small dataset is sufficient to capture the dynamics. Conversely, for a more diverse task group, the prediction error of the world model should decrease as the volume of training data increases.

The results are shown in Table 2. For GenSim, the model evaluation error for group *Piles* (where the objects of interest are piles of pellets) is significantly lower than others. This aligns with the fact that all tasks in this group only involve pushing piles on the table to a specified location. On the other hand, *Placement*, involves placing different types of objects in different manners, showing a much

Table 3: Imitation learning performance on different projects. GenSim reports step-wise rewards with 1.0 indicating success. RoboGen reports raw rewards which may have arbitrary scales, which we convert to success rate for intuitive understanding.

| Method | GenSim | | | | RoboGen | | BBSEA | |
|---|---|---|---|---|---|---|---|---|
| Task Group | Stacking | Placement | Piles | Assembling | Table-Top | Ground | Table-Top | Drawer |
| Reward/ Success Rate | 0.15 | 0.32 | 0.2 | 0.08 | 0.0 | 0.0 | 0.00 | 0.00 |

model higher error when trained on a small number of trajectories. Regarding BBSEA, cases are similar to those in GenSim. However, for RoboGen, both groups exhibit low model errors. This is primarily due to RoboGen's learning design: for each task, all trajectories start with the same initial state. Therefore, the dynamics seen by the agent are similar and easy to learn by the world model.

### 4.3 GENERALIZATION

We train a Diffusion Policy for each task group using 40 trajectories, identical to the data for dynamics model training, and assess their performance on the same task group under variations such as scene configurations, object colors, and initial robot states. As indicated in Table 3, GenSim demonstrates reasonable generalization performance, despite the challenge posed by randomizing object colors, which complicates the effectiveness of an RGB-based policy. Conversely, agents trained on RoboGen and BBSEA tasks exhibit poor generalization. For RoboGen, the primary issue is that the training trajectories all begin from the same initial state, and the RL solutions do not generate high-quality data. In the case of BBSEA, the problem often lies in the repetition of similar tasks, which restricts task-level generalization capabilities. Moreover, significant task variations can result in out-of-distribution challenges that adversely affect agent performance.

## 5 CONCLUSION AND DISCUSSION

In this paper, we propose a novel evaluation framework for generative simulation, which includes three fundamental metrics: quality, diversity and generalization. We evaluate three representative generative simulation pipelines based on our proposed method. Results indicate that while various pipelines excel in terms of quality and diversity, there remains significant potential for improvement in their generalization capabilities. We hope that future work in generative simulation can make advancements and improvements in these three areas, especially in terms of generalization.

Moreover, we identify and outline some common drawbacks and failure cases across current generative simulation pipelines as follows for instructions to encourage further exploration:

- **Low-quality task descriptions:** Although task proposal is not a bottleneck for generative simulation in general, we still observe some vague and repeated task descriptions that fail to express the details of the generated tasks. Such ambiguity may cause suboptimal results in the evaluation of text-based task diversity, as well as harm the performance of a language-conditioned policy.

- **Trajectory data with limited diversity:** The task solution in some methods only considers limited task and scene variations, which will affect both the trajectory diversity and task-level generalization capability. Typical cases include insufficient intra-task randomization, relatively fixed task domain, or identical semantics between different tasks, leading to very similar trajectories. We advocate an appropriate dynamics model trained along with the task generation process to inspect and improve the diversity regarding dynamics coverage in future works.

- **Task-specific design data collection and imitation learning:** We remark that, on designing the task generation pipeline, generalization could be considered and improved in various ways, e.g., action space with good abstraction (control by end-effector poses, joint positions, or primitive actions), data augmentation, and unified goal specification. For example, in GenSim, actions are abstracted as high-level waypoints, and each task trajectory contains only a few such high-level actions. This design benefits its generalization evaluation based on imitation learning. We aim to devise a generally applicable protocol, with the diffusion policy *not tuned or adopted specifically*, and advocate efficient domain-specific designs for better generalization performance.

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

# A APPENDIX

## A.1 PROMPT FOR EVALUATION

Here is the prompt for the evaluation of the completion score:

```
You are an assessment expert responsible for the Task Completion rate.
    Your task is to score the completion rate for the task in the
    following rules:

        1. Evaluate the completion rate for the robotics task.
        2. During the evaluation, you will receive 8 images and a basic
            description of the task.
        3. During the evaluation, you need to make very careful judgments
            and evaluate the completion of the task based on the order
            of the pictures and the task description.
        4. In the evaluation you need to pay attention to the smoothness
            of the trajectory.
        5. Assign a score between 0 and 10, with 10 being the highest. Do
            not provide a complete answer.
        6. Your should provide the answer in the following format:

    Score: X
```

Here is the prompt for the evaluation of the scene alignment score without the caption model:

```
You are an assessment expert responsible for Task description and Scene
    images pairs. Your task is to score the Scene caption according to
    the following requirements:

    1. Evaluate how well the Scene images covers the scene of the
        robotics task. You should consider whether the scene is similar
        to the requirement of the task.
    2. During the evaluation, you will receive 4 images and a basic
        description of the task.
    3. In the evaluation, you should pay attention to the alignment
        between the Scene image and the real-world task following the
        description of the task.
    4. A good scene should not only provide an environment for completing
         a robotics task but should also contain items that may appear
        near the task, even though they may have nothing to do with the
        task itself.
    5. Assign a score between 1 and 5, with 5 being the highest.
    6. Your should provide the answer in the following format:

Score: X
```

Here is the prompt for the evaluation of the scene alignment score with the caption model:

```
You are an assessment expert responsible for Task description and Scene
    captions pairs. Your task is to score the Scene caption according to
    the following requirements:

    1. Evaluate how well the Scene captions covers the scene of the
        robotics task. You should consider whether the scene is similar
        to the requirement of the task.
    2. In the evaluation, you should pay attention to the alignment
        between the Scene captions and the real-world task following the
        description of the task.
    3. A good scene should not only provide an environment for completing
         a robotics task but should also contain items that may appear
```

```
      near the task, even though they may have nothing to do with the
      task itself.
  4. Scene caption sets will be a set of different views to the scene.
  5. Assign a score between 1 and 5, with 5 being the highest. Do not
      provide a complete answer; give the score in the format: Score: 3
```

## A.2 HUMAN EVALUATION RESULT

In table 4 and table 5, we collected the scoring results from 18 researchers in the field of robotics on 20 tasks, which were sourced from RoboGen and GenSim, respectively. We conducted five evaluations for each model, with the values in parentheses representing the variance of the five evaluations. Based on these two tables, we derived the results presented in Figure 3.

Table 4: Comparison of Human Evaluation and Various Models on Scene Alignment Score of Different Tasks.

| Task Name (RoboGen) | Human | GPT-4+Blip2 | Cambrian | LLava | GPT-4v | GPT-4+LLava | GPT-4+Cambrian |
|---|---|---|---|---|---|---|---|
| Open Laptop | 3.6 | 2.56(0.9) | 2.00(0) | 3.00(0) | 3.96(0.03) | 3.20(1.20) | 2.00(0.00) |
| Change Lamp Direction | 3.45 | 2.88(0.75) | 2.00(0) | 3.00(0) | 3.96(0.03) | 3.80(0.08) | 3.28(0.21) |
| Flush the Toilet | 3.55 | 3.00(0.91) | 0.00(0) | 3.00(0) | 3.40(0.02) | 2.80(1.20) | 4.00(0) |
| Extend Display Screen | 2.55 | 2.16(0.13) | 0.00(0) | 4.00(0) | 3.80(0.14) | 2.00(0) | 1.96(0.01) |
| Close the Oven Door | 3.45 | 1.80(0.2) | 0.00(0) | 3.00(0) | 3.60(0.04) | 2.00(0) | 2.00(0) |
| Set Oven Timer | 3.20 | 2.80(1.2) | 0.00(0) | 3.00(0) | 3.04(0.03) | 3.96(0.01) | 4.00(0) |
| Close Window | 3.10 | 2.40(0.48) | 1.00(0) | 3.00(0) | 3.96(0.01) | 1.60(0.30) | 1.00(0) |
| Adjust Water Flow | 2.85 | 1.40(0.3) | 2.00(0) | 3.00(0) | 3.36(0.03) | 1.24(0.11) | 1.48(0.05) |
| Open Both Table Doors | 3.20 | 2.24(0.29) | 5.00(0) | 3.00(0) | 3.24(0.05) | 2.00(0) | 2.00(0) |
| Press Start Button | 3.20 | 3.24(1.29) | 0.00(0) | 3.00(0) | 3.48(0.03) | 2.20(2.70) | 4.00(0) |
| **Task Name (Gensim)** | **Human** | **GPT-4+Blip2** | **Cambrian** | **LLava** | **GPT-4v** | **GPT-4+LLava** | **GPT-4+Cambrian** |
| Align Balls in Colored Boxes | 4.30 | 4.00(0.00) | 2.00(0) | 3.00(0) | 3.36(0.07) | 1.33(0.33) | 2.00(0) |
| Block Pyramid with Limited Space | 4.10 | 4.00(3.00) | 2.00(0) | 3.00(0) | 3.64(0.07) | 1.50(0.50) | 2.04(0.01) |
| Align Spheres in Colored Zones | 4.20 | 4.40(0.16) | 2.00(0) | 3.40(0.30) | 4.32(0.03) | 2.00(0.00) | 2.00(0) |
| Color Coded Blocks on Corner | 3.15 | 4.13(0.05) | 0.40(0.80) | 3.00(0) | 3.68(0.07) | 2.00(0) | 3.24(0.61) |
| Align Rope Cross Zone | 4.35 | 3.53(0.65) | 2.00(0) | 1.00(0) | 3.56(0.11) | 1.00(0) | 1.80(0.20) |
| Color Ordered Insertion | 4.40 | 2.00(0.00) | 2.00(0) | 3.00(0) | 4.08(0.11) | 2.00(0) | 2.00(0) |
| Color Specific Container Fill | 4.25 | 2.00(0.00) | 0.60(0.30) | 3.00(0) | 3.84(0.03) | 2.00(0) | 2.20(0.20) |
| Color Coordinated Zone Stacking | 3.70 | 4.27(0.21) | 2.00(0) | 3.00(0) | 3.88(0.03) | 1.00(0) | 2.40(0.80) |
| Vertical Insertion Blocks | 3.75 | 3.13(1.77) | 1.60(0.80) | 3.00(0) | 3.44(0.09) | 2.07(1.21) | 2.20(0.20) |
| Color Blocks in Cylinder Maze | 2.65 | 2.47(0.65) | 0.00(0) | 3.00(0) | 3.12(0.05) | 2.00(0) | 1.76(0.13) |

## A.3 DIVERSITY AND GENERALIZATION EXPERIMENT DETAILS

### A.3.1 TASK SELECTION AND GROUPING

For GenSim, we use all the templates and generated tasks released by the authors. For RoboGen, we only use the manipulation tasks but not locomotion and soft body because the locomotion tasks yield very poor learning performance and the soft-body tasks are not publicly available at the time. For BBSEA, we perform generation following the instructions provided by the authors.

We group these tasks mainly for two reasons: (1) grouped tasks offer more perspectives for analysis, and (2) the latent dynamics model and diffusion policy training, with their original implementation, are insufficient for learning a large number of tasks. The dimensions to consider include scene

Table 5: Comparison of Human Evaluation and Various Models on Completion Score of Different Tasks

| Task Name (RoboGen) | Human | Cambrian-8B | LLava-1.5 | GPT-4 |
|---|---|---|---|---|
| Open Laptop | 8.2 | 0.00(0) | 8.00(0) | 7.96(0.01) |
| Change Lamp Direction | 8.4 | 0.00(0) | 8.00(0) | 6.40(0.92) |
| Flush the Toilet | 8.3 | 0.00(0) | 8.00(0) | 7.40(0.06) |
| Extend Display Screen | 5.1 | 0.00(0) | 8.00(0) | 6.36(1.11) |
| Close the Oven Door | 4.8 | 0.00(0) | 8.00(0) | 5.80(1.02) |
| Set Oven Timer | 7.8 | 0.00(0) | 8.00(0) | 6.40(0.04) |
| Close Window | 9.3 | 0.00(0) | 8.00(0) | 6.80(0.42) |
| Adjust Water Flow | 7.2 | 0.00(0) | 8.00(0) | 7.64(0.07) |
| Open Both Table Doors | 6.3 | 0.00(0) | 8.00(0) | 5.76(1.35) |
| Press Start Button | 8.4 | 0.00(0) | 8.00(0) | 6.88(0.81) |
| **Task Name (Gensim)** | **Human** | **Cambrian-8B** | **LLava-1.5** | **GPT-4** |
| Align Balls in Colored Boxes | 9.2 | 0.00(0) | 8.00(0) | 4.92(0.23) |
| Block Pyramid with Limited Space | 6.8 | 0.00(0) | 8.00(0) | 5.16(2.35) |
| Align Spheres in Colored Zones | 4.9 | 0.00(0) | 8.00(0) | 3.16(2.97) |
| Color Coded Blocks on Corner | 6.2 | 0.00(0) | 8.00(0) | 6.48(1.27) |
| Align Rope Cross Zone | 8.7 | 0.80(3.20) | 8.00(0) | 4.25(2.81) |
| Color Ordered Insertion | 9.5 | 1.60(4.80) | 8.00(0) | 5.56(1.21) |
| Color Specific Container Fill | 9.2 | 0.80(3.20) | 8.00(0) | 5.88(0.43) |
| Color Coordinated Zone Stacking | 8.4 | 0.00(0) | 8.00(0) | 5.52(0.83) |
| Vertical Insertion Blocks | 8.1 | 0.00(0) | 8.00(0) | 3.72(1.31) |
| Color Blocks in Cylinder Maze | 6.3 | 0.00(0) | 8.00(0) | 6.08(1.69) |

configuration (e.g., table-top vs. ground), skill, and objects involved (e.g., pick-and-place using a suction gripper vs. moving piles of grains using a shovel-like end-effector).

A summary with examples is shown in Table 6.

Table 6: Details of Task Grouping

| Project | Group | # Tasks | Desc. Examples |
|---|---|---|---|
| GenSim | Placement | 19 | "cylinder-line-placement: place cylinders of different colors on a line at specific location" |
| | Stacking | 30 | "stack-cylinder-on-bowl: stack cylinders of matching colors on top of bowls" |
| | Piles | 11 | "sweeping-piles: push piles of small objects into a target goal zone" |
| | Assembling | 14 | "build-bridge: construct a bridge using two yellow blocks and three blue blocks" |
| RoboGen | Table-Top | 17 | "Adjust Water Flow: the robotic arm will turn one of the faucets hinge switches to adjust the flow of the water" |
| | Ground | 15 | "Adjust Chair Position: the robot arm will adjust the position of the unfolded chair" |
| BBSEA | Table | 49 | "Gather all objects and organize them in the green bin" |
| | Drawer | 26 | "Open the drawer using the drawer handle" |

### A.3.2 TRAJECTORY COLLECTION

Trajectories are collected for both dynamics model learning and imitation learning. GenSim implements an oracle agent for generating demonstrations. The oracle agent's action specifies the target end-effector pose command, which is executed by a low-level Inverse Kinematics controller with

joint commands. Therefore, we collect transitions of both high-level (target end-effector pose as actions) and low-level (joint commands as actions) with the observations being RBG image and robot joint states. We collect for each task 40 trajectories with manually varied random seeds.

RoboGen decomposes a long-horizon task into multiple stages, each solved by motion planning or reinforcement learning. In this work, we are only concerned with the sub-tasks that require reinforcement learning. Specifically, we run their pipeline to train a policy for each task and subsequently collect trajectories using that policy. The action space includes joint and gripper commands and the observation space includes RGB images and robot joint states. We collect for each task 40 trajectories with manually varied random seeds.

BBSEA generates trajectory demonstration by querying ChatGPT to output parameterized action primitives, which are then executed by low-level controllers. Since BBSEA does not have officially released tasks, we run the pipeline for generation and collection for each of the 32 scenes, giving 256 trajectories in total. BBSEA's proposed pipeline additionally filters success trajectories. But here we use all trajectories for learning the dynamics model.

### A.3.3 DYNAMICS MODEL TRAINING DETAILS

We adopt a popular community implementation (Hafner et al., 2024). For all experiments, the model is trained for 10 epochs with a batch size of 8. The data was chunked into sequences of size 40/40/20 for GenSim/RoboGen/BBSEA. All other hyperparameters are kept as default. Since DreamerV3 is designed for reinforcement learning from visual observations, its model architecture is expressive and robust to different domains. Data augmentation could be used to improve its robustness to aspects such as the variation in color and appearance further to obtain a lower prediction error. However, we do not incorporate that in this paper for simplicity.

### A.3.4 IMITATION LEARNING MODEL TRAINING DETAILS

For GenSim and RoboGen, the implementation is adapted from the official release of Diffusion Policy (Chi et al., 2023). We use the configuration provided by the authors of Diffusion Policy for image-state observation. For all experiments, the policy is trained for 8000 epochs. For BBSEA, we use the image-language diffusion policy implementation provided by the authors.

