# OpenReview forum: "On Evaluation of Generative Robotic Simulations"
_ICLR.cc/2025/Conference — Submitted to ICLR 2025_

### Official Review · Reviewer_XFSC · 2024-10-30

**Soundness:** 2
**Presentation:** 2
**Contribution:** 2
**Rating:** 5
**Confidence:** 3

**Summary:**

This paper addresses the problem of evaluating autonomously generated tasks by VLM in simulation. The authors propose an evaluation strategy focusing on three aspects: the quality of individual generated tasks, the diversity of multiple generated tasks, and the generalization ability of a diffusion policy trained on the generated tasks. The approach is applied to tasks generated by RoboGen, GenSim, and BBSEA, and the evaluation scores are compared with human evaluation scores.

**Strengths:**

* The paper tackles an important and under-explored problem: evaluating the quality of generated tasks in a systematic way.
* The criteria used for evaluation—task quality, task diversity, and generalization ability—are all crucial for assessing whether a generated task set is useful.

**Weaknesses:**

* **Clarity of contribution**: The contribution is not entirely clear. Is this the first evaluation strategy for task sets? If not, how does it compare to existing evaluation methods, such as Self-BLEU or Embedding similarity scores from RoboGen? More clarification is needed on how this approach improves upon or differs from existing evaluation techniques.
* **Realism of generated tasks**: The claim regarding "evaluating the realism of generated tasks" seems overreaching. The approach measures scene alignment score, which reflects consistency between task descriptions and scene images. However, it's not clear why this metric necessarily reflects the "realism" of the generated tasks. A more thorough justification or evidence supporting this claim would strengthen the paper.
* **Human evaluation details**: The paper lacks specifics on the human evaluation process. Human evaluations were conducted on 10 generated tasks, but key details are missing. How many human evaluators were involved? Were they experts in robotic tasks or general participants? What guidelines were provided to ensure consistency in ratings, particularly for scene alignment and task completion scores?
* **Writing and presentation**: The paper could benefit from improved clarity in writing, particularly in explaining key concepts and the methodology used.

**Questions:**

1. **Pearson correlation in Figure 3**: Figure 3 shows Pearson correlation between evaluation methods and human evaluations. However, it does not quantify how much the proposed evaluation method deviates from human evaluation. Why are the scores in Figure 3 (left) generally better than those in Figure 4 (right)? Does this imply that the proposed method is more reliable for scene alignment scoring than for task completion scoring?
2. **Figure 4 method clarification**: The paper mentions various LLM/VLM options in Section 4.1.1, but it’s unclear which one is used in Section 4.1.2 Figure 4\. Do all VLM/LLM choices yield similar results, or do specific models shows the result trend in Figure 4? For example, does RoboGen consistently outperform in task completion scores while GenSim excels in scene alignment, regardless the choice of evaluation VLM/LLM?
3. **Line 354 clarification**: The paper attributes the shortfall in performance to the vision-language model’s “lack of access to top-view rendered data”. Is this limitation specific to the VLM used in GenSim or to the VLM used in the proposed evaluation method?
4. **Task set generalization (Section 4.3)**: In this section, different task groups are used to measure the generalization ability (as shown in the Appendix Table 6). Given that the task groups are not identical across task sets, is it fair to conclude that GenSim exhibits better generalization while the others perform poorly? How does this variability in the definition of task groups affect the generalization conclusions?
5. **Definition of text embedding (Line 264\)**: What is the formal definition of the text embedding e\_i​ used for task descriptions? More clarity here would help.
6. **Image sampling (Line 246\)**: The paper mentions extracting 8 images from the trajectory video. Are these images randomly sampled, or are they evenly spaced throughout the trajectory? Does the selection method include the final frame? Could different sampling strategies affect the task completion score?

---

> ### Author Response · Authors · 2024-11-21
>
> We sincerely appreciate your comprehensive evaluation and feedback on our paper.
>
> > Clarity of contribution: The contribution is not entirely clear. Is this the first evaluation strategy for task sets? If not, how does it compare to existing evaluation methods, such as Self-BLEU or Embedding similarity scores from RoboGen? More clarification is needed on how this approach improves upon or differs from existing evaluation techniques.
>
> We agree that the issue you raised is indeed important. First, different generative simulation works use entirely distinct evaluation methods. For example, Robogen evaluates diversity using Self-BLEU and Embedding Similarity, while BBSEA and Gensim assess the quality of tasks generated by different large language models (LLMs). In contrast, we propose a more comprehensive and complete evaluation method suitable for all generative simulation tasks, allowing for a more accurate and appropriate assessment of these approaches' capabilities, rather than relying solely on text-level diversity comparisons among generated tasks. Our evaluation extends beyond text-level diversity, assessing factors such as task quality, the diversity of generated task trajectories, and the applicability of generated tasks in imitation learning. Therefore, we believe our work represents the first complete evaluation framework for generative simulation.
>
> > Realism of generated tasks: The claim regarding "evaluating the realism of generated tasks" seems overreaching. The approach measures scene alignment score, which reflects consistency between task descriptions and scene images. However, it's not clear why this metric necessarily reflects the "realism" of the generated tasks. A more thorough justification or evidence supporting this claim would strengthen the paper.
>
> Regarding "evaluating the realism of generated tasks," we acknowledge that our wording may have been imprecise. Our primary focus is on the consistency between the scene and the text. While we aim for the scenes to be reasonably realistic, realism itself is not the central focus of our evaluation.
>
> > How many human evaluators were involved? Were they experts in robotic tasks or general participants? What guidelines were provided to ensure consistency in ratings, particularly for scene alignment and task completion scores?
>
> We had 18 participants in the evaluation, primarily researchers in robotics(expert in robotic tasks). In the questionnaire, we provided guidelines on how to score correctly, so we believe we offered sufficient instructions.
>
> > Writing and presentation: The paper could benefit from improved clarity in writing, particularly in explaining key concepts and the methodology used.
>
> We will refine the wording of our paper based on the reviewers' feedback.
>
>
>
> > Pearson correlation in Figure 3: Figure 3 shows Pearson correlation between evaluation methods and human evaluations. However, it does not quantify how much the proposed evaluation method deviates from human evaluation. Why are the scores in Figure 3 (left) generally better than those in Figure 4 (right)? Does this imply that the proposed method is more reliable for scene alignment scoring than for task completion scoring?
>
>
> The Pearson correlation score is lower which means the evaluation method deviates from human evaluation. Scene alignment requires less capability from the model. In contrast, completion demands a higher level of skill, such as the ability to understand relationships between multiple images, which typically necessitates more advanced models to achieve better results in completion tasks. Therefore, we believe that as large model capabilities continue to improve, the gap in addressing these issues will be resolved.
>
> > Figure 4 method clarification: The paper mentions various LLM/VLM options in Section 4.1.1, but it’s unclear which one is used in Section 4.1.2 Figure 4. Do all VLM/LLM choices yield similar results, or do specific models shows the result trend in Figure 4? For example, does RoboGen consistently outperform in task completion scores while GenSim excels in scene alignment, regardless the choice of evaluation VLM/LLM?
>
> We used GPT-4V, as it demonstrated the best performance in Figure 3.
>
> > Line 354 clarification: The paper attributes the shortfall in performance to the vision-language model’s “lack of access to top-view rendered data”. Is this limitation specific to the VLM used in GenSim or to the VLM used in the proposed evaluation method?
>
> The limitation here arises because Gensim tasks are typically presented in a top-down view, while visual-language models like LLaVa and Cambrian generally lack similar top-down view data for robotics. As a result, their performance on Gensim is relatively weaker.

---

> ### Author Response · Authors · 2024-11-21
>
> > Task set generalization (Section 4.3): In this section, different task groups are used to measure the generalization ability (as shown in the Appendix Table 6). Given that the task groups are not identical across task sets, is it fair to conclude that GenSim exhibits better generalization while the others perform poorly? How does this variability in the definition of task groups affect the generalization conclusions?
>
> The reason we opt for grouping the tasks is the limited capability of vanilla Diffusion Policy to handle a multi-task setting. We ackowledge that task spliting would quantitatively affect generalization results.
>
> |        | Assembling | Piles | Placement | Stacking |
> |--------|------------|-------|-----------|----------|
> | Reward | 0.1        | 0.21  | 0.6       | 0        |
>
> > Definition of text embedding (Line 264): What is the formal definition of the text embedding $e_i$ used for task descriptions? More clarity here would help.
>
> In this context, a formal definition for the text embedding $e_i$ would clarify its purpose, structure, and how it is generated. Generally, text embeddings are vector representations of text that capture semantic meaning and allow comparisons across different textual data. Here’s an example of a precise definition:
>
> Let $e_i$ be the embedding of the task description $T_i$, where $T_i$ is a sequence of tokens representing the description. The embedding $e_i$ is a vector in $\mathbb{R}^d$ that is generated by applying a text embedding model $f$, such as a pretrained language model (e.g., BERT, GPT, etc.), to $T_i$. Formally, this can be written as: $e_i = f(T_i) \in \mathbb{R}^d$
>
> where $f$ is a mapping function that converts the tokenized text $T_i$ into a fixed-dimensional vector representation in $\mathbb{R}^d$. The dimension $d$ depends on the embedding model used and is typically chosen to balance representational power and computational efficiency.
>
>
>
> > Image sampling (Line 246): The paper mentions extracting 8 images from the trajectory video. Are these images randomly sampled, or are they evenly spaced throughout the trajectory? Does the selection method include the final frame? Could different sampling strategies affect the task completion score?
>
> We uniformly extract 8 images from each video, including the final frame. Different sampling methods can affect our results. For instance, in the "Open The Laptop" task, we applied the following sampling methods:
>
> 1. Uniform sampling
> 2. Random sampling
> 3. Selected sampling(Random sampling with a focus on the latter half of the video)
> 4. Sampling without the final frame
>
> The results are as follows:
>
> | Metric                          | Human Evaluation | Random Sampling | Selected Sampling | Sampling without final frame | Ours |
> |---------------------------------|------------------|-----------------|-------------------|-----------------------------|------|
> | Score                           | 8.2              | 7.6             | 4.0              | 8.0                         | 8.0  |
>
> Thus, we believe our sampling method is superior to random and selective random sampling. However, since sampling has a relatively small impact on the results, we did not conduct large-scale experiments on sampling methods.

---

> ### Author Response · Authors · 2024-11-24
>
> Dear Reviewer XFSC,
>
> Thank you for taking the time to review our work and provide your feedback. We hope our responses have adequately addressed your concerns. If you have any further questions or wish to discuss any aspects of the paper, we would be glad to engage during the remaining discussion period.
>
> On the other hand, if you feel that your concerns have been fully resolved, we kindly request that you consider reevaluating your rating. Your support means a great deal to us and would contribute significantly to the recognition of our work.
>
> Thank you!

---

> > ### Comment · Reviewer_XFSC · 2024-11-25
> >
> > Thank you for the detailed response! I'm still a bit concerned about the contribution. Are the proposed metrics sufficient to measure these three aspects? Are the proposed three aspects complete enough to evaluate the generated simulation platform? Are there any other factors not considered? (such as whether the policy learned on the generated environment has a high potential of transferring to the real world? whether the difficulty distribution of generated tasks is proper for policy learning?)  Therefore, I'd like to keep my rating.

---

> > > ### Author Response · Authors · 2024-11-26
> > >
> > > We believe the three aspects we proposed are sufficient to evaluate the capabilities of generative simulation, and our evaluation metrics comprehensively cover these three aspects. Could you clarify where you think there might be gaps in coverage or areas insufficient for evaluating generative simulation capabilities?
> > >
> > > Regarding the sim-to-real issue you raised, as mentioned in our response to another reviewer, current models lack sufficient generalization capabilities to solve tasks in the real world. If we were to define a sim-to-real setting ourselves, we believe our evaluation standard would lose its fairness. Therefore, we did not include a sim-to-real evaluation.
> > >
> > > As for the difficulty of tasks, if your concern is whether tasks are excessively difficult to the point that no policy can solve them, the completion score reflects whether tasks are overly challenging. If you are concerned that the policies trained in our generalization experiments are too difficult to solve, we provide the success rates of our methods on the training set in the table below to address this concern.
> > >
> > > Full results of generalization. **Train** indicates evaluation on task configurations that are generated with the same seeds as in the dataset, while  **Eval** uses configurations from new seeds. We varified that diffusion policy is powerful enough to fit to the training dataset with a very small imitation loss/error (1e-4). Note that however, GenSim only ensures that the initial states are identical with the same seed but not the object colors, making the distribution slightly drifts from that of the dataset.
> > >
> > >  As shown in the table, the learned policy achieves good performance on Placement bucause Placement usually involves short-horizon and a few objects, e.g., putting a block into a container, making generalization easy. Piles is harder because the dynamics of the piles (little grains scattered on the table) are complex. Placement and Stacking are very hard because they are usually long-horizon and involves reasoning of relations between objects. And the effect of comounding error is significant: the learned policy can often execute the first step, but fail to complete the subsequent steps accurately, resulting in final failures.
> > >
> > >
> > > |                | Assembling | Piles | Placement | Stacking |
> > > |----------------|------------|-------|-----------|----------|
> > > | Reward (Train) | 0.21       | 0.76  | 0.8       | 0.1      |
> > > | Reward (Eval)  | 0.1        | 0.21  | 0.6       | 0        |

---

> > > ### Author Response · Authors · 2024-12-02
> > >
> > > Dear Reviewer XFSC,
> > >
> > > Thank you for taking the time to review our work and provide your feedback. We hope our responses have adequately addressed your concerns. If you have any further questions or wish to discuss any aspects of the paper, we would be glad to engage during the remaining discussion period.
> > >
> > > On the other hand, if you feel that your concerns have been fully resolved, we kindly request that you consider reevaluating your rating. Your support means a great deal to us and would contribute significantly to the recognition of our work.
> > >
> > > Thank you!

---

### Official Review · Reviewer_2PAM · 2024-11-02

**Soundness:** 2
**Presentation:** 3
**Contribution:** 3
**Rating:** 3
**Confidence:** 4

**Summary:**

This paper presents an evaluation framework for simulations generated by the generative foundational model, focusing on three key metrics: (i) the realism of tasks and generated trajectories, (ii) task diversity, and (iii) generalization to new tasks. This framework indicates a correlation between expert human evaluations.

**Strengths:**

- This paper seeks to tackle significant challenges in robotics, especially the high costs associated with evaluating generative simulations. The proposed framework leverages the capabilities of LLMs/VLMs to automate this process.
- Certain VLM/LLM choices demonstrate a correlation with human expert evaluations of task quality, suggesting the potential effectiveness of these methods. However, some VLM/LLM show less alignment with human assessments.

**Weaknesses:**

- **Clarification on Experiment Design**
  + Below are some points for clarifying the experiment setup:
  + Could the authors explain why the three selected metrics are considered essential for generative simulation? While single-task quality and task diversity are understandable, it's unclear how generalization fits within this context. How does it differ from simply having diverse tasks?
  + Could the authors clarify how scene alignment is evaluated? From my understanding of lines 348-350, it seems to rely on the robot's trajectory. If that’s the case, shouldn’t the quality of the generated trajectory also play a role? Additionally, I’m curious why scene alignment is based on video rather than image.
  + The connection between high prediction loss and task diversity is unclear. For instance, a single task could involve varied dynamics, like table rearrangement. Could the authors provide a justification for this?

- **Fairness of Evaluation**
  + Each baseline uses a different task for evaluation, but certain tasks or simulator setups (as mentioned in lines 318-319) might inherently pose challenges for LLM/VLM-based score evaluation, potentially affecting fair comparison.

**Questions:**

- How do humans evaluate the task? (e.g., Do they use a 1-5 scoring system, rank tasks by preference, or something else?)
- Related to the point above (in the Weaknesses section, high prediction loss), lines 274-277 weren’t clear about the exploration aspect. Could the authors clarify how diverse dynamics are connected to task diversity?

### Minor Suggestions
- Section 3.3 on Scene Alignment Score could reference prompt (A.1).
- Section 4.1.1 could refer to the detailed experiment setup in (A.2).

---

> ### Author Response · Authors · 2024-11-21
>
> Thank you very much for thoroughly reviewing our work and providing constructive feedback.
>
> > Could the authors explain why the three selected metrics are considered essential for generative simulation? While single-task quality and task diversity are understandable, it's unclear how generalization fits within this context. How does it differ from simply having diverse tasks?
>
> Our work primarily focuses on proposing an evaluation method for generative simulation that reflects the overall quality of the generated data. In generative simulation, enabling the data to serve general-purpose robots is a fundamental goal, making generalization a crucial evaluation criterion. Generalization is distinct from diversity: text diversity merely reflects the range of tasks generative simulation could potentially produce, while trajectory diversity reflects whether a single task can generate a sufficient variety of complex trajectories. Generalization, on the other hand, emphasizes that the generated data should effectively support imitation learning, enabling models to learn the ability to solve diverse tasks.
>
>
> > Could the authors clarify how scene alignment is evaluated? From my understanding of lines 348-350, it seems to rely on the robot's trajectory. If that’s the case, shouldn’t the quality of the generated trajectory also play a role? Additionally, I’m curious why scene alignment is based on video rather than image.
>
>
> We did not use trajectory videos to evaluate scene alignment. Instead, we used multi-view images from different perspectives when evaluating Robogen and BBSEA, while for Gensim, we used top-down view images of the scenes. We will clarify the evaluation method in main content.
>
>
> > The connection between high prediction loss and task diversity is unclear. For instance, a single task could involve varied dynamics, like table rearrangement. Could the authors provide a justification for this?
>
> The predictive capability of the same model after the same training duration is consistent. Therefore, if the model shows a higher loss on different datasets, it suggests that the dataset contains trajectories and scenes with greater diversity. Thus, we believe that high prediction loss can indicate higher trajectory and visual diversity. Additionally, if a task is highly complex, it will also yield a relatively high prediction loss, further suggesting that a dataset containing such a task has greater diversity.
>
> > Each baseline uses a different task for evaluation, but certain tasks or simulator setups (as mentioned in lines 318-319) might inherently pose challenges for LLM/VLM-based score evaluation, potentially affecting fair comparison.
>
> The tasks generated by different generative simulation pipelines are inconsistent, so we chose multiple similar tasks within the same domain for evaluation. It is not feasible to have all generative simulation pipelines generate identical tasks in the same simulation for evaluation, as the content of the tasks they produce is entirely different.
>
> > How do humans evaluate the task? (e.g., Do they use a 1-5 scoring system, rank tasks by preference, or something else?)
>
> Human experts used a scoring system of $1–5$ and $1–10$ for rating.
>
> > Related to the point above (in the Weaknesses section, high prediction loss), lines 274-277 weren’t clear about the exploration aspect. Could the authors clarify how diverse dynamics are connected to task diversity?
>
> Same to the response above.

---

> ### Author Response · Authors · 2024-11-24
>
> Dear Reviewer 2PAM,
>
> Thank you for taking the time to review our work and provide your feedback. We hope our responses have adequately addressed your concerns. If you have any further questions or wish to discuss any aspects of the paper, we would be glad to engage during the remaining discussion period.
>
> On the other hand, if you feel that your concerns have been fully resolved, we kindly request that you consider reevaluating your rating. Your support means a great deal to us and would contribute significantly to the recognition of our work.
>
> Thank you!

---

> ### Comment · Reviewer_2PAM · 2024-11-25
> **Clarification on the project contribution.**
>
> Thank you authors, for providing the clarifications.
>
> However, I am still a bit unclear about the main contribution of this work. While it seems to be motivated by the idea of a "generative simulator" that could support the application of multiple algorithms (e.g., IL, RL), I would appreciate it if the authors could elaborate on what is meant by "generative simulation that reflects the overall quality of the generated data." Specifically, is the primary focus on the "data" itself or on the simulation environment?
>
> Additionally, I found the statement "the same model after the same training duration is consistent" a bit unclear, and I am not entirely convinced by the claim that "high prediction loss can indicate higher trajectory and visual diversity." High prediction loss could also result from the inherent complexity of the dynamics (e.g., contact-rich scenarios), and it’s not clear how this directly relates to visual diversity.
>
> I would be happy to see further clarifications from the authors.

---

> > ### Author Response · Authors · 2024-11-25
> >
> > > However, I am still a bit unclear about the main contribution of this work. While it seems to be motivated by the idea of a "generative simulator" that could support the application of multiple algorithms (e.g., IL, RL), I would appreciate it if the authors could elaborate on what is meant by "generative simulation that reflects the overall quality of the generated data." Specifically, is the primary focus on the "data" itself or on the simulation environment?
> >
> > It is important first to clarify the "Generative Simulation" concept, which refers to a pipeline that generates robotics tasks within a simulation environment using foundational models. Our evaluation of "Generative Simulation" focuses on assessing the quality of the tasks produced by this pipeline. To achieve this, we first need to evaluate the overall quality, diversity, and generalization capability of the generated "dataset." Based on this comprehensive evaluation of the dataset, we can then infer the task generation ability of the "Generative Simulation" pipeline, i.e., the "overall quality of the generated data." The quality of the simulation is not directly considered in the evaluation pipeline.
> >
> > >Additionally, I found the statement "the same model after the same training duration is consistent" a bit unclear, and I am not entirely convinced by the claim that "high prediction loss can indicate higher trajectory and visual diversity." High prediction loss could also result from the inherent complexity of the dynamics (e.g., contact-rich scenarios), and it’s not clear how this directly relates to visual diversity.
> >
> > We believe that for a world model, the prediction loss when forecasting the next image after training on similar datasets for the same duration can reflect the diversity of video trajectories at the image level within the dataset. For example, for the same task, the image-level diversity of videos generated from different trajectories can be well-captured. Similarly, for different tasks, variations in environments can also introduce a certain degree of diversity at the image level.
> >
> > Therefore, we interpret the prediction loss from the world model as an indicator of the dataset's ability to generate diverse trajectories across different environments. A higher prediction loss suggests better diversity in both environments and trajectories. We refer to this metric as "dynamics diversity," which represents the diversity of robot behaviors and environments within the dataset.
> >
> > If any part remains unclear, we welcome further questions.

---

> > ### Author Response · Authors · 2024-12-02
> >
> > Dear Reviewer 2PAM,
> >
> > Thank you for taking the time to review our work and provide your feedback. We hope our responses have adequately addressed your concerns. If you have any further questions or wish to discuss any aspects of the paper, we would be glad to engage during the remaining discussion period.
> >
> > On the other hand, if you feel that your concerns have been fully resolved, we kindly request that you consider reevaluating your rating. Your support means a great deal to us and would contribute significantly to the recognition of our work.
> >
> > Thank you!

---

### Official Review · Reviewer_MwfY · 2024-11-03

**Soundness:** 3
**Presentation:** 3
**Contribution:** 3
**Rating:** 8
**Confidence:** 3

**Summary:**

This work presents an evaluation framework for "generative" robot simulators. Here, generative simulations are a simulator, task set, and possibly set of demonstrations for each task. The authors propose evaluating these simulations in three criteria: quality, diversity, and generalization. Task quality is evaluated by scene alignment between the task description and the scene, and also scoring the task completion video with an mLLM to see whether the associated data or policy completes the task. For task diversity, the authors evaluate the description embedding, and also a trajectory prediction error to understand the data/trajectory diversity. Finally, for generalization, the authors train policy on the training data and evaluate – seeing how well the performance is on evaluation set.

**Strengths:**

1. This paper is pretty timely and addresses an important recent development in simulation based robotics – generative simulations. While there are multiple such works coming out in recent months, there is no easy way to understand which one someone should use for their applications. A quantitative evaluation like the one proposed in the paper goes a long way to address this concern.

2. The three axes proposed by the authors are also reasonable – and the way to evaluate them also seems reasonable. Splitting the three criteria into the smaller sub-criteria is also pretty pertinent.

3. The fact that the authors also do human evaluations to find alighment between the mLLM evaluations and human evaluations make their evaluation more trustworthy.

4. The task diversity evaluations are also interesting – although maybe less meaningful than the other axes since the type of tasks i.e the task class proposed by each simulator may vary a lot.

5. Finally, the authors expose interesting issues with the generalization ability in RoboGen and BBSEA – that the data generated by them is not high quality, and thus they need to improve their data quality before they can be useful.

**Weaknesses:**

1. There doesn't seem to be a consistent correlation with human evaluation and mLLM proposed results in Figure 3.
2. The alignment score given by the mLLMs is not calibrated – some calibration would make the results more reasonable.
3. Generalization evaluation would be more convincing if it was trained on some kind of task-conditioned multi-task policy like BAKU [1] rather than an unconditional policy like diffusion policy.

[1] Haldar, Siddhant, Zhuoran Peng, and Lerrel Pinto. "BAKU: An Efficient Transformer for Multi-Task Policy Learning." arXiv preprint arXiv:2406.07539 (2024).

**Questions:**

1. Do the authors think the task diversity captures the overall size of the possible task sets in a generative simulation?
2. Would the authors be open sourcing their evaluation suite so developers of generative simulations can develop with this benchmark in mind?

---

> ### Author Response · Authors · 2024-11-21
>
> We genuinely thank you for your thorough assessment and encouraging remarks regarding our manuscript.
>
> > There doesn't seem to be a consistent correlation with human evaluation and mLLM proposed results in Figure 3.
>
>
> Although in Figure 3, most models do not perform well, GPT-4V shows some alignment with human expert ratings in certain areas, such as scene alignment in Gensim and completion score in Robogen. We acknowledge that current large foundation models are not yet capable of adequately evaluating scene alignment and task completion scores. However, we believe that future models will better align with human expert evaluations in these areas.
>
> > The alignment score given by the mLLMs is not calibrated – some calibration would make the results more reasonable.
>
> Thank you very much for raising this point. We did not perform calibration for each large model; however, by calculating the correlation with human behavior, we minimized issues arising from the lack of calibration. Therefore, we believe our conclusions remain reasonable. That said, adding calibration would indeed make our conclusions more convincing.
>
> Based on our understanding of calibration, we modified the evaluation prompts by adding both positive and negative examples. For the BLIP2+GPT4 method, this was achieved by modifying the prompts alone, while for GPT4V, we incorporated additional images along with the prompts. The table below presents the experimental results after incorporating calibration. Due to time constraints, we conducted these tests only on the Robogen dataset as a reference.
>
> | Model                     | GPT-4v + BLIP2 | Only GPT-4v | GPT4 + LLAVA | GPT4V + Calibration | GPT4 + BLIP2 + Calibration |
> |---------------------------|----------------|--------------|-----------------|----------------------|---------------------------|
> | **Score**                 | 0.574179       | 0.205204    | 0.557254     | 0.220560             | 0.317414                  |
>
>
> > Generalization evaluation would be more convincing if it was trained on some kind of task-conditioned multi-task policy like BAKU [1] rather than an unconditional policy like diffusion policy.
>
> Thank you very much for your suggestion. We agree that BAKU is a suitable model for imitation learning; however, it was only made public three months ago, so we did not use it as a benchmark when completing our work. Although BAKU meets our needs, we believe it is reasonable to choose the more widely known diffusion policy method for our evaluation. If future work demonstrates that BAKU can effectively assess data generalization, we will add related experiments to our codebase for use by future researchers. We will discuss this work in related work.
>
> > Do the authors think the task diversity captures the overall size of the possible task sets in a generative simulation?
>
> If "overall size of the possible task sets" refers to the number of non-semantically repetitive tasks that can be generated, then higher text-based diversity would likely result in more such tasks. However, if you are referring to tasks with different trajectories, we believe that simply running the generative simulation pipeline continuously can produce an unlimited number of tasks with varying trajectories.
>
> > Would the authors be open sourcing their evaluation suite so developers of generative simulations can develop with this benchmark in mind?
>
> Yes. We will release the code in the camera-ready version.

---

> ### Author Response · Authors · 2024-11-24
>
> Dear Reviewer MwfY,
>
> Thank you for taking the time to review our work and provide your feedback. We hope our responses have adequately addressed your concerns. If you have any further questions or wish to discuss any aspects of the paper, we would be glad to engage during the remaining discussion period.
>
> On the other hand, if you feel that your concerns have been fully resolved, we kindly request that you consider reevaluating your rating. Your support means a great deal to us and would contribute significantly to the recognition of our work.
>
> Thank you!

---

> > ### Comment · Reviewer_MwfY · 2024-11-25
> > **Thank you for your response**
> >
> > Dear authors, thanks for responding to my queries. I still believe that some kind of task conditioned behavior cloning (even older architectures like MT-ACT or RT-1) would have made the experiments better. I appreciate your update on calibrating the scores. Overall, I am happy with the (rather high) score I posted on my initial review, which was conditioned on satisfactory answers from your end.

---

> > > ### Author Response · Authors · 2024-11-26
> > >
> > > We sincerely appreciate your positive feedback and recognition of our work. Your insightful comments have been invaluable in helping us further enhance our submission. Thank you for your thoughtful and constructive input.

---

### Official Review · Reviewer_MyJX · 2024-11-04

**Soundness:** 2
**Presentation:** 2
**Contribution:** 2
**Rating:** 3
**Confidence:** 4

**Summary:**

This paper proposes a framework to analyze existing simulation benchmarks for embodied AI. The authors consider 3 metrics: quality, diversity, and generalization. For quality, authors analyze the realism of the generated task and the completeness of the generated trajectories. For diversity, they measure instruction diversity and trajectory diversity. For generalization, they train a diffusion policy and check success rate on unseen tasks. The results reveal that while metrics of quality and diversity can be achieved through certain methods, no single approach excels across all metrics.

**Strengths:**

1. The authors proposes 3 aspects (quality, diversity, and generalization) to assess current simulation benchmarks.
2. Experiments and analysis are interesting.

**Weaknesses:**

1. I think in addition to quality, diversity, and generalization, sim-to-real gap is a big issue that needs to be addressed/discussed when analyzing simulation benchmark. Have you considered metrics to assess how well the simulated tasks translate to real-world scenarios? Or maybe include a section on potential sim-to-real challenges for each evaluated benchmark.
2. It seems the diversity score only considers the text and trajectory. However, I think it should also consider at least visual input.
3. It seems this paper did a great job of analysis of existing benchmarks, but did not propose a new benchmark to resolve these issues.
4. It seems generalization and diversity has some connection. The diversity is to measure how out-of-distribution tasks are from each other while generalization is to measure how in-distribution tasks are with each other. I wonder whether we can use the same metric to measure these 2 things. Is it possible to investigate correlation between diversity and generalization scores? Or is it possible to potentially unify these two metrics?

**Questions:**

See above.

---

> ### Author Response · Authors · 2024-11-21
>
> Thank you for your thoughtful evaluation.
>
> > Have you considered metrics to assess how well the simulated tasks translate to real-world scenarios?
>
> We agree that sim-to-real gap is an important problem should be considered. While assessing the transfer from simulation to the real world is indeed important, there is no universally accepted real-world setting. If we were to define our own sim-to-real transfer approach, it would compromise the fairness of the evaluation. Therefore, we believe that evaluating sim-to-real cannot be conducted in a fair and reasonable manner. Also we think that many of our metrics aim to minimize the sim-to-real gap.
>
> > It seems the diversity score only considers the text and trajectory. However, I think it should also consider at least visual input.
>
> Visual diversity is important for training policies in imitation learning; however, it provides limited insight into the quality of the data itself and is not closely aligned with the aspects we aim to evaluate. Achieving visual diversity in a simulator can be relatively simple by adding data augmentation; for example, adding complex but irrelevant textures to the simulator background may quickly increase visual diversity without actually improving the quality or diversity of the generated data. Additionally, within trajectory diversity, the dynamics model also captures some basic visual diversity—tasks with relatively high visual diversity often exhibit greater trajectory diversity as well.
>
>
> Visual diversity is indeed an important aspect and is actually already captured by the visual reconstruction part of the dynamics error. We further evaluate the visual diversity of Gensim through the LPIPS score and the image reconstruction part of Dreamer's loss function:
>
>
> |            | Assembling | Piles | Placement | Stacking |
> |------------|------------|-------|-----------|----------|
> | LPIPS Mean | 12.2       | 11.1  | 15.9      | 13.9     |
> | LPIPS Std  | 11.7       | 10.7  | 15.5      | 13.5     |
> | Image Loss | 16.8       | 8.0   | 14.8      | 15.9     |
>
> where a lower Mean/Std indicates more similar and hence less diverse tasks. The results align with our dynamics error measure that Piles is the least diverse and Placement is the most diverse.
>
>
>
> >It seems this paper did a great job of analysis of existing benchmarks, but did not propose a new benchmark to resolve these issues.
>
> Our work aims to establish a unified evaluation standard that future work in the domain of generative simulation can build upon to achieve improvements by addressing the issues we highlight. We believe that developing a robust evaluation benchmark is more important than simply outperforming existing metrics. Many evaluation studies also did not propose better benchmarks aligned with their evaluations ~[1,2,3], which is why, in this work, we did not introduce a new generative simulation pipeline.
>
> [1] Papineni K, Roukos S, Ward T, et al. Bleu: a method for automatic evaluation of machine translation[C]//Proceedings of the 40th annual meeting of the Association for Computational Linguistics. 2002: 311-318.
>
> [2] He Y, Bai Y, Lin M, et al. T $^ 3$ Bench: Benchmarking Current Progress in Text-to-3D Generation[J]. arXiv preprint arXiv:2310.02977, 2023.
>
> [3] Hu Y, Li T, Lu Q, et al. Omnimedvqa: A new large-scale comprehensive evaluation benchmark for medical lvlm[C]//Proceedings of the IEEE/CVF Conference on Computer Vision and Pattern Recognition. 2024: 22170-22183.

---

> ### Author Response · Authors · 2024-11-21
>
> > It seems generalization and diversity has some connection. The diversity is to measure how out-of-distribution tasks are from each other while generalization is to measure how in-distribution tasks are with each other. I wonder whether we can use the same metric to measure these 2 things. Is it possible to investigate correlation between diversity and generalization scores? Or is it possible to potentially unify these two metrics?
>
> 'Generalization' is distinct from 'Diversity' under the context of generative simulation. Text diversity merely reflects the range of potential tasks that generative simulation can produce, while trajectory diversity indicates whether a single task can generate sufficiently varied and complex trajectories. In contrast, generalization emphasizes that the generated data should effectively support imitation learning, enabling models to learn the capability to handle different tasks. Though diversity and generalization can be related, they are neither identical nor unified. As shown in Tables 1, 2, and 3 of the paper, although BBSEA and RoboGen outperform GenSim in diversity, their generalization capabilities are not as strong as those of GenSim. From a qualitative perspective,both extremely low and high levels of diversity can undermine generalization capabilities. Considering a scenario where a dataset consists of a single task trajectory, representing minimal diversity, the policy will inevitably overfit, resulting in no generalization. Conversely, if data diversity is excessively high to the extent that the model or algorithm cannot effectively capture and learn from it (e.g. the case in RoboGen), generalization will also suffer due to a decline in overall training performance. This highlights the need for a balanced approach to diversity in training data to optimize generalization.

---

> ### Author Response · Authors · 2024-11-24
>
> Dear Reviewer MyJX,
>
> Thank you for taking the time to review our work and provide your feedback. We hope our responses have adequately addressed your concerns. If you have any further questions or wish to discuss any aspects of the paper, we would be glad to engage during the remaining discussion period.
>
> On the other hand, if you feel that your concerns have been fully resolved, we kindly request that you consider reevaluating your rating. Your support means a great deal to us and would contribute significantly to the recognition of our work.
>
> Thank you!

---

> > ### Comment · Reviewer_MyJX · 2024-11-26
> >
> > Thanks for the reply.
> > However, my concerns remain for several aspects.
> >
> > 1. For sim-to-real, I acknowledge that it is hard, but trying to avoid the hard problem will lead to misleading conclusions. Back in old days, there are a bunch of RL algorithms that can do super well in simulator, but a lot of these algorithms turn out to not work very well in real world settings. Therefore, understand sim-to-real and have a good metrics or have a way to know whether a benchmark considers that part is a critical question to the community.
> >
> > 2. I did not quite get the visual diversity evaluation. Would you elaborate?
> >
> > 3. re: proposing new benchmark, it is a lot of work and challenging to propose a new benchmark. However, I think this is the bar for an ICLR paper.
> >
> > 4. re: generalization vs diversity. It sounds like you are using "diversity" to measure the how out-of-distribution it is for the training set, while you use "generalization" to measure how out-of-distribution the model can handle during evaluation? Is it correct?

---

> > > ### Author Response · Authors · 2024-11-26
> > >
> > > > For sim-to-real, I acknowledge that it is hard, but trying to avoid the hard problem will lead to misleading conclusions. Back in old days, there are a bunch of RL algorithms that can do super well in simulator, but a lot of these algorithms turn out to not work very well in real world settings. Therefore, understand sim-to-real and have a good metrics or have a way to know whether a benchmark considers that part is a critical question to the community.
> > >
> > > First, we acknowledge that sim-to-real is important. However, it is crucial to clarify what kind of sim-to-real the community truly needs. While it is possible to reproduce simulator scenarios perfectly in the real world to achieve partial task success using the learned policy, we believe such sim-to-real approaches should not be encouraged. Instead, sim-to-real methods that deserve evaluation are those where models with sufficient generalization capabilities can solve tasks in the real world even when the real-world environment differs from the simulator. This type of sim-to-real is what we believe should be advocated.
> > >
> > > As shown in the paper, even datasets like Gensim lack sufficient generalization capabilities to enable models to solve real-world problems. Therefore, at this stage, we consider it inappropriate to propose sim-to-real evaluations, as the current datasets lack the generalization needed to learn policies capable of solving real-world tasks.
> > >
> > > Additionally, since the community does not have a widely accepted sim-to-real approach, defining one ourselves and using it as a basis for evaluating datasets' sim-to-real capabilities would be unfair. After carefully considering sim-to-real evaluation, we decided not to use it to assess the capabilities of "generative simulation." Our goal is to provide a fair and effective evaluation method for "generative simulation," rather than relying on techniques to transfer policies learned in the simulator to the real world.
> > >
> > > > I did not quite get the visual diversity evaluation. Would you elaborate?
> > >
> > > We apologize for the in-adequate explanations regarding visual diversity evaluation. Here are the details:
> > >
> > > We first uniformly sample N=2000 images from the dataset for each task group. For each group, the images are passed into a pretrained network (in this case [1]) and compute their LPIPS [2]. LPIPS is pair-wise a similarity score where a low value indicates high similarity. Averaging over the N sampled images, both high mean and variance values indicate high low similarity and high diversity. The results are shown in the first two rows in the table.
> > >
> > > The second evaluation is through error of the dynamics model [3]:
> > >
> > > $$
> > > L(\phi) = E [-\ln p_\phi(x_t| z_t, h_t)  -\ln p_\phi (r_t|z_t, h_t) -\ln p_\phi (c_t|z_t, h_t)]
> > > $$
> > >
> > > where $p_\phi$ is a decoder that tries to reconstruct the visual input $x_t$, reward $r_t$ from latent states $z_t$ and $h_t$. Here we are concerned with the visual part $-\ln p_\phi(x_t| z_t, h_t)$ for visual diversity. A high reconstruction error/loss indicates high visual diversity. We evaluate the final dynamics models trained on 40 episodes per task to compute this error for each task group. The result is shown in row 3 of the table.
> > >
> > > Both approaches give suggest the task group Piles is relatively less diverse while Placement is the more diverse, aligning with our conclusion in the paper and thus validating our method.
> > >
> > > [1] Hu, Jie, Li Shen, and Gang Sun. "Squeeze-and-excitation networks." Proceedings of the IEEE conference on computer vision and pattern recognition. 2018.
> > > [2] Zhang, Richard, et al. "The unreasonable effectiveness of deep features as a perceptual metric." Proceedings of the IEEE conference on computer vision and pattern recognition. 2018.
> > > [3] Hafner, Danijar, et al. "Mastering diverse domains through world models." arXiv preprint arXiv:2301.04104 (2023).

---

> > > ### Author Response · Authors · 2024-11-26
> > >
> > > > re: proposing new benchmark, it is a lot of work and challenging to propose a new benchmark. However, I think this is the bar for an ICLR paper.
> > >
> > > As mentioned in our rebuttal, developing a new benchmark may indeed be an important contribution and could be a suitable submission to ICLR. However, for the community, a comprehensive evaluation method is even more critical. Our proposed evaluation framework provides clear guidance for researchers attempting to create new benchmarks, helping them better understand how to improve the capabilities of generative simulation. We believe that a well-designed evaluation framework also merits submission to ICLR. Just as past researchers have proposed evaluation standards in their works [4,5,6], we have clearly outlined a method for evaluating "Generative Simulation" in our paper. We will present a new benchmark after.
> > >
> > > [4] Papineni K, Roukos S, Ward T, et al. Bleu: a method for automatic evaluation of machine translation[C]//Proceedings of the 40th annual meeting of the Association for Computational Linguistics. 2002: 311-318.
> > >
> > > [5] He Y, Bai Y, Lin M, et al. T $^ 3$ Bench: Benchmarking Current Progress in Text-to-3D Generation[J]. arXiv preprint arXiv:2310.02977, 2023.
> > >
> > > [6] Hu Y, Li T, Lu Q, et al. Omnimedvqa: A new large-scale comprehensive evaluation benchmark for medical lvlm[C]//Proceedings of the IEEE/CVF Conference on Computer Vision and Pattern Recognition. 2024: 22170-2218
> > >
> > > > re: generalization vs diversity. It sounds like you are using "diversity" to measure the how out-of-distribution it is for the training set, while you use "generalization" to measure how out-of-distribution the model can handle during evaluation? Is it correct?
> > >
> > > Our evaluation of generalization focuses on in-distribution rather than out-of-distribution performance, even though we split the dataset into training and testing sets. For diversity evaluation, we assess the diversity across the entire dataset, including both the training and testing sets, rather than limiting the evaluation to the diversity of the training set alone.m
> > >
> > > Yes, you are right. But a better way to put is that, for diversity, we measure **how much space can be covered** by the distribution of the training dataset. If the coverage is large, i.e., diverse, the relative "out-of-distribution-ness" for each individual sample is indeed larger as you have point out. For generalization, we examine **how well do the generated data promote model generalization on these tasks**. Bad designs, such as inappropriate observation/action/episode design, that prevent the model from handling tasks from the same distribution well, should be captured by a poor generalization performance during evaluation.

---

> > > ### Author Response · Authors · 2024-12-02
> > >
> > > Dear Reviewer MyJX,
> > >
> > > Thank you for taking the time to review our work and provide your feedback. We hope our responses have adequately addressed your concerns. If you have any further questions or wish to discuss any aspects of the paper, we would be glad to engage during the remaining discussion period.
> > >
> > > On the other hand, if you feel that your concerns have been fully resolved, we kindly request that you consider reevaluating your rating. Your support means a great deal to us and would contribute significantly to the recognition of our work.
> > >
> > > Thank you!

---

### Author Response · Authors · 2024-11-21
**General Response**

We thank all reviewers and ACs for their time and effort in reviewing the paper. We are glad that the reviewers generally recognized the following contributions. All reviewers have recognized that our work proposes a novel evaluation framework for Generative Simulation. Some reviewers **found our study on the consistency with human evaluation behavior to be convincing** (`MwfY`,`2PAM`,`XFSC`). Additionally, regarding the three aspects we proposed, some reviewers expressed agreement, **acknowledging our contributions in defining and evaluating these different axes**. (`MwfY`,`XFSC`)

### Key Answers
Several reviewers have raised similar concerns. We address them below.

***How is 'Generalization' different from 'Diversity':*** Our work focuses on proposing an evaluation method for generative simulation that reflects the overall quality of the generated data. **For generative simulation, enabling data to serve a broad range of robotic tasks is a fundamental objective, making generalization a critical evaluation criterion**. Unlike diversity, which merely indicates the range of potential tasks (text diversity) or the complexity and variety of trajectories for a single task (trajectory diversity), generalization emphasizes the need for generated data to support imitation learning. In other words, diversity measures how diverse the generated data are, while generalization reflects how generally the trained policy performs. (`2PAM`) Though diversity and generalization can be related, they are neither identical nor unified. As shown in Tables 1, 2, and 3 of the paper, although BBSEA and RoboGen outperform GenSim in diversity, their generalization capabilities are not as strong as those of GenSim. From a qualitative perspective,both extremely low and high levels of diversity can undermine generalization capabilities. Considering a scenario where a dataset consists of a single task trajectory, representing minimal diversity, the policy will inevitably overfit, resulting in no generalization. Conversely, if data diversity is excessively high to the extent that the model or algorithm cannot effectively capture and learn from it (e.g. the case in RoboGen), generalization will also suffer due to a decline in overall training performance. This highlights the need for a balanced approach to diversity in training data to optimize generalization. (`MyJX`)

***How to evaluate 'Scene Alignment':*** For complex scenes, we used multi-view images for evaluation, instead of robot trajectories, as we believe that assessing a scene from a single view does not accurately reflect the complexity of such scenarios. (`2PAM`) Additionally, we observed a degree of consistency between the model's and human performance in scene alignment. (`MwfY`) Regarding "the realism of generated tasks", it includes whether the scene is aligned with the text, as well as the “semantics” of real scenarios. Failure examples of the former case include simulation penetration, or putting something supposed to be in kitchen into a bathroom. We have updated the paper for clarification. (`XFSC`)

***Task Diversity:*** Visual diversity is important for training policies in imitation learning, but it provides limited insight into the intrinsic quality of the data. The relevance of visual diversity as a metric does not closely align with the aspects we aim to evaluate. Achieving visual diversity in a simulator can be straightforward; for example, adding complex but irrelevant textures to the background of a simulator might artificially increase visual diversity without actually improving data quality or diversity. (`MyJX`) Additionally, in trajectory diversity, the dynamics model often captures some basic visual diversity; tasks with relatively high visual diversity tend to exhibit greater trajectory diversity. Similarly, if a single task includes sufficiently varied policies, it also indicates good trajectory diversity. (`2PAM`)

***Open Source:*** We will open source our evaluation pipeline in the camera-ready version.

---

### Meta-Review · Area_Chair_xZmF · 2024-12-26

**Metareview:**

The authors of this paper take on a very challenging task: analyzing the quality, diversity, and generalization of various robotics simulators. They evaluate these based on a mixture of different methods, using large language models for quality, world-model predictions for diversity, and zero-shot generalization of trained policies for generalization. They compare to human evaluations.

Strengths:
Given the importance of robotics simulators, it's important to study their qualities and come up with metrics to assess them. This paper seems like a good and interesting first step in that direction.

Weaknesses:
No characterization of sim-to-real or physical realism of a simulation
Characterization of visual diversity is lacking
Only two simulations, GenSim and RoboGen - why not Robocasa, for example?
Generalization metric is not task-conditioned -- which actually would punish simulators for creating "hard" environments where multiple tasks are possible.
Some of the language model

While the authors raised very important points about the quality of the simulators, I think there are a large number of clarifications still necessary, and the writing of the paper could be improved, especially if they want others to use the metrics they have proposed.

**Additional Comments On Reviewer Discussion:**

Because this paper is proposing metrics by which we can study simulations, it's almost a position paper, and therefore many of the comments read as suggestions or discussion points and are not especially technical. This is part of the challenge of writing a paper like this one.

For example, MyJX raised a number of points about the semantics of terms the authors used (quality, diversity, and generalization), and the authors responded in detail about why not to include things like sim-to-real. If they cannot do sim-to-real, it seems visual diversity would be an important metric to assess, in order to have it as a proxy.

Reviewer Mwfy pointed out some of the mLLM scores for quality were not well aligned with human ratings, and raised a point about task-conditioned policies for generalization.

Explanations about things like visual diversity (MyJX) were missing,.

XFSC raised concerns about the realism of generated tasks which mirror MyJX's concerns about sim-to-real. Generative robotics simulators are, ideally, supposed to let us train policies that are useful for robots. Instead, the authors look for consistency between scene and text. This is a fairly big difference, in my opinion.

Many details about human evaluation were also only provided in comments in the rebuttal and should be added to the paper.

Due to concerns about the contributions, many reviewers (eg XFSC) kept their scores the same. I think that, particularly for a paper whose main contribution must be compelling, persuasive arguments about metrics, this is a huge issue. We want to convince people that these metrics are useful when designing new generative simulations.

As a result, I ended up weighting many of these reviews, even though I feel like some of the concerns reviewers raised - especially about creating new benchmarks from MyJX -- were unnecessary and unfair.

---

### Decision · Program_Chairs · 2025-01-22

Reject